# *RGS4* RNA Secondary Structure Mediates Staufen2 RNP Assembly in Neurons

**DOI:** 10.3390/ijms222313021

**Published:** 2021-12-01

**Authors:** Sandra M. Fernández-Moya, Janina Ehses, Karl E. Bauer, Rico Schieweck, Anob M. Chakrabarti, Flora C. Y. Lee, Christin Illig, Nicholas M. Luscombe, Max Harner, Jernej Ule, Michael A. Kiebler

**Affiliations:** 1Department for Cell Biology & Anatomy, Biomedical Center (BMC), Medical Faculty, Ludwig-Maximilians-University, 82152 Planegg-Martinsried, Germany; sandra.fernandez@bmc.med.lmu.de (S.M.F.-M.); Janina.Ehses@bmc.med.lmu.de (J.E.); karl.bauer@bmc.med.lmu.de (K.E.B.); rico.schieweck@bmc.med.lmu.de (R.S.); christin.illig@bmc.med.lmu.de (C.I.); max.harner@bmc.med.lmu.de (M.H.); 2RNA Network Laboratory, The Francis Crick Institute, 1 Midland Road, London NW1 1AT, UK; anob.chakrabarti@crick.ac.uk (A.M.C.); flora.lee@crick.ac.uk (F.C.Y.L.); nicholas.luscombe@crick.ac.uk (N.M.L.); jernej.ule@crick.ac.uk (J.U.); 3Department of Neuromuscular Diseases, UCL Queen Square Institute of Neurology, Queen Square, London WC1N 3BG, UK; 4Department of Genetics, Environment and Evolution, UCL Genetics Institute, London WC1E 6BT, UK; 5Genomics and Regulatory Systems Unit, Okinawa Institute of Science & Technology Graduate University, 1919-1 Tancha, Onna-son, Kunigami-gun, Okinawa 904-0495, Japan; 6Department of Molecular Biology and Nanobiotechnology, National Institute of Chemistry, SI-1001 Ljubljana, Slovenia

**Keywords:** Staufen2, neuronal RNA granule, RNP assembly, RNA-binding protein, RNA localization, in vivo RNA binding

## Abstract

RNA-binding proteins (RBPs) act as posttranscriptional regulators controlling the fate of target mRNAs. Unraveling how RNAs are recognized by RBPs and in turn are assembled into neuronal RNA granules is therefore key to understanding the underlying mechanism. While RNA sequence elements have been extensively characterized, the functional impact of RNA secondary structures is only recently being explored. Here, we show that Staufen2 binds complex, long-ranged RNA hairpins in the 3′-untranslated region (UTR) of its targets. These structures are involved in the assembly of Staufen2 into RNA granules. Furthermore, we provide direct evidence that a defined *Rgs4* RNA duplex regulates Staufen2-dependent RNA localization to distal dendrites. Importantly, disrupting the RNA hairpin impairs the observed effects. Finally, we show that these secondary structures differently affect protein expression in neurons. In conclusion, our data reveal the importance of RNA secondary structure in regulating RNA granule assembly, localization and eventually translation. It is therefore tempting to speculate that secondary structures represent an important code for cells to control the intracellular fate of their mRNAs.

## 1. Introduction

RNA-binding proteins (RBPs) are instrumental in assembling RNA into ribonucleoprotein particles (RNPs, often also called RNA granules) [1,2], thereby mediating spatio-temporally controlled regulation of gene expression, which is particularly relevant in neurons [3]. This asymmetric RNA localization is now known to be much more common than previously assumed, occurring in many different cell types and organisms [4,5]. Moreover, several hundreds of proteins with potential RNA binding ability have been identified [6]. This opens the question how cells orchestrate the RBP network in order to control the transcriptome and translatome. There is increasing evidence that RNA secondary structures critically contribute to regulation of the fate of mRNAs. Recently, RNA structure has been shown to allow mRNAs to self-associate and to promote RNA condensation into RNA liquid droplets [7,8,9]. Furthermore, RNA structure critically contributes to mRNA transport [10,11]. In mammalian neurons, many different RBPs have been implicated in RNA localization [4,12], among them two RBPs with a proven function in RNA transport: ZBP1/Igf2bp1 and the double-stranded RBP Staufen2 (Stau2) [5]. Stau2 and its paralog Stau1 are both crucial regulators of neuronal function [13]. Even though it has been shown that both proteins are involved in the Up-frameshift 1 mediated RNA decay pathway [14], they control different pathways within neurons [13,15] suggesting that Stau1 and Stau2 regulate distinct RNA targets.

The nerve cell-enriched Stau2 is needed for mRNA localization in primary neurons and the brain [16,17,18]. Therefore, Staufen proteins preferentially bind within the 3′-untranslated region (UTR) of their target mRNAs to regulate their expression in neurons [19,20]. Apparently, defined elements characterized by a complex secondary structure therein are important for binding of Staufen proteins to their targets [21,22]. Previous studies have reported in vitro binding of Staufen proteins to certain transcripts and defined RNA structures in the 3′-UTR [23,24]. However, data on physiologically relevant Stau2 binding sites in mammalian cells or in vivo are still lacking. In addition, Stau1 and Stau2 seem to have non-redundant functions in nerve cells [13] indicating that they display a preference for distinct RNA structures within the transcriptome. Therefore, we set out to identify specific RNA secondary structures that are recognized by Stau2 using hiCLIP (RNA hybrid and individual-nucleotide resolution ultraviolet crosslinking and immunoprecipitation). In addition to the identification of target transcripts, hiCLIP provides structural information on the bound RNA which is needed to model complex structures (i.e., [25]). In the current study, we succeeded in functionally characterizing the role of specific RNA hairpins in RNP assembly, dendritic mRNA transport and control of protein expression in primary neurons.

## 2. Results

### 2.1. Stau2 Binds RNA Structures in the 3′-UTR In Vivo

In order to gain insight into Stau2 binding to mRNAs in the brain and to identify specific RNA structures within its targets, we have performed hiCLIP [26] for Stau2 from rat brain at day 35 (P35), during the period of synaptic maturation (Figure 1A). The hiCLIP approach thereby allows the unraveling of specific RNA secondary structures that interact with a specific double-stranded RBP such as Stau2 in vivo (as shown for Stau1; [26]). We performed gene-level enrichment analysis of Stau2 immuno-precipitates (IPs) compared to input control in order to identify Stau2-bound mRNAs. This led to the selective enrichment of Stau2 targets, such as the Microtubule-associated protein tau (*Mapt*), the Ras homolog family member A (*RhoA*) or *Rgs4* (Figure 1B). First, we compared the newly identified Stau2 targets with previously published Stau2 IP data from embryonic brains [19]. This allowed us to identify a core of 181 common Stau2 targets out of 811 postnatally detected mRNAs (Figure 1C). Next, we performed functional Gene Ontology (GO) term analysis with the core Stau2 targets revealing the terms synaptic regulation and plasticity (Appendix A). Interestingly, we observed similar GO terms when we compared Stau2 targets that localize in the neuropil (Appendix A). Amongst them was the previously identified Stau2 target, *Rgs4* (Figure 1B, highlighted in red), together with a number of known dendritically localized targets, nicely confirming the validity of our hiCLIP approach.

To identify key targets directly bound and regulated by Stau2 during synaptic maturation, we compared the significantly enriched mRNA duplexes from postnatal brains (P35) with previously published Stau2-dependent up- and downregulated mRNAs from embryonic brains (E17) [19]. As Stau2 clearly contributes to RNA localization in neurons [15,18,20] and in the brain [16], we also referred to a previously published dataset on localized mRNAs from the neuropil layer of hippocampal area CA1 [27]. From those two data sets, i.e., neurons and brain, 359 Stau2 RNA targets were preferentially localized in the neuropil layer of the CA1 region from hippocampal slices (Figure 1D). The fact that there is only limited overlap between the different data sets provides important new insight into the complexity of the process studied. Interestingly also for this restricted data set, functional GO term analysis revealed the terms synaptic regulation and plasticity (Appendix A). Of particular interest to us was the overlap of 14 target mRNAs (highlighted in dark grey, including *Rgs4*), between all three data sets. Together, this encouraged us to proceed with our subsequent analysis to determine Stau2 binding and identify specific RNA secondary structures within the Stau2 target mRNAs.

To identify biologically relevant RNA targets, we filtered our dataset for the following three criteria (Figure 1D): (i) does Stau2 bind to the respective 3′-UTR in rat brain cortex (P35), (ii) is a target known to be regulated by Stau2 and finally (iii) does it regulate synaptic function? Rewardingly, *Rgs4* mRNA is one of the top-scoring Stau2 targets (criteria 1) that we previously started to characterize in neurons [18,19]. Interestingly, *Rgs4* is also regulated by Stau2 ([19], criteria 2). The mRNA of *Rgs4* encodes a GTPase-activating protein of the G protein-coupled receptor (GPCR) pathway important for synaptic function ([28,29], criteria 3). Several members of this pathway including *Rgs4*, *Calm3* and *Gna11* are bound by Stau2 (Figure 1D) [19]. Therefore, we focused on *Rgs4* mRNA to study the role of RNA secondary structures in Stau2 mediated regulation. In addition, we chose *RhoA* and *Mapt* as important controls, both being important localized mRNAs in primary neurons (highlighted in Figure 1B, criteria 1).

### 2.2. A Defined Long-Range RNA Structure Recruits Stau2

The increased resolution of our hiCLIP approach reveals the presence of complex RNA structures within *Rgs4*, mainly in the beginning of the 3′-UTR (Figure 2A,B). Notably, the conserved RNA structure in panel A (Figure 2A, black box) forms an RNA duplex over long distance, with the two strands of the stem (orange boxes) separated by 167 nts (represented by the light blue arc). This newly identified RNA duplex is close to a previously predicted structure identified as a Staufen Recognition Site (SRS3, compare the light and dark blue duplex arc shown in Figure 2A) [21]. Interestingly, in silico folding (RNAfold, ViennaRNA) of the *Rgs4* fragment F1 did not predict the novel Stau2 bound RNA structure identified by hiCLIP (compare Figure 2B with Appendix A). The novel hiCLIP structure depicted in Figure 2B could only be calculated in silico by prior fusion of the two arms of the long-range hairpin detected by hiCLIP. This highlights the importance of the Stau2 hiCLIP approach in vivo. To maximize the effect of the selected Stau2 RNA structure, we subcloned the beginning of the 2.2 kb *Rgs4* 3′-UTR (*Rgs4* fragment 1/*Rgs4* F1, see Materials for details, Figure 2A). To evaluate whether the identified RNA structure within the *Rgs4* 3′-UTR is indeed responsible for the assembly of Stau2 containing particles in distal dendrites of mature (14 DIV) hippocampal neurons, we generated a 20 nt deletion (*F1Δ*) to disrupt the Stau2 bound RNA duplex (Figure 2A, black box, Figure 2B, black line indicates the deletion). To visually assay dendritic RNA localization, we used the well-established MS2 reporter system in neurons [30], already optimized in our lab for *Rgs4* [18]. In total, four constructs (*MS2*-control, *Rgs4* 3′-UTR full length/*Rgs4* FL, *Rgs4* F1, or *Rgs4 F1Δ*) were tested for Stau2 recruitment by co-expressing a red fluorescently tagged version of the longest isoform (62 kDa) of the Stau2 protein, tagRFP-Stau2 (Figure 2C). Co-localization of MS2 reporter RNA and tagRFP-Stau2 was calculated as fraction of MS2 RNA particles containing Stau2 (Figure 2D and Appendix A). Both *MS2-Rgs4 FL* and *MS2-Rgs4 F1* constructs showed a significantly higher number of co-localization events (48% ± 3% and 44% ± 1%, respectively) between tag-RFP-Stau2 (in magenta) and MS2 RNA particles (in green) compared to *MS2-*control RNA. Importantly, disruption of the Stau2 RNA duplex in *MS2-Rgs4 F1Δ*) led to a significant reduction of co-localization to control levels (32% ± 1% and 31% ± 1%, respectively) (Figure 2D,E). Together, these results are consistent with the idea that disrupting the *Rgs4* RNA duplex prevents Stau2 from associating with the reporter RNA, which in turn would reduce the number of Stau2-*Rgs4* containing particles in dendrites.

### 2.3. RNA Structure Conveys Stau2-Dependent Dendritic mRNA Localization

As RNA structure critically contributes to RNP assembly both under physiological as well as pathological conditions [31], we set out to analyze the role of the Stau2 RNA duplex in dendritic mRNA localization. For this, we once again used the MS2 reporter system in hippocampal neurons (Figure 3A). All MS2 reporter constructs (*Rgs4* FL, *Rgs4* F1, *Rgs4* F1Δ, control) were found to localize to distal dendrites to varying degrees (Figure 3B–D and Appendix A). To assess dendritic localization quantitatively, we chose two different parameters. First, the average distance of dendritic particles from the cell body (Figure 3B, or displayed in 40 µm segments in Appendix A) and second a density plot displaying the distribution of RNA granules along the dendrite (Figure 3C). Importantly, there is no obvious differences in the localization of the different constructs under normal conditions. One possible explanation could be that Stau2 might also bind the control constructs (Figure 2D). Upon Stau2 depletion, however, dendritic mRNA localization—dependent on the presence of the *Rgs4* structure—is significantly affected (Figure 3B–D and Appendix A). This is clearly seen in the shift of dendritic RNA granules towards the cell body, especially for *Rgs4* F1 and, to a lesser extent, for *Rgs4* FL (Figure 3C, colored areas). Together, these findings provide experimental evidence that the identified RNA structure—and its interaction with Stau2—is needed for distal localization of *Rgs4* mRNA within dendrites. We then asked whether the absence of Stau2 would reduce the overall levels of dendritic RNA compared to control conditions as previously reported in [20]. Consistent with this idea, the total amount of RNA particles detected in Stau2 knockdown conditions was significantly smaller than that observed in control neurons (Figure 3D and Appendix A), providing strong evidence that Stau2 is important for the delivery of RNA to dendrites. In conclusion, our data argue for a functional role of the newly identified RNA duplex serving as the driving force for Stau2-dependent assembly of long, complex folded 3′-UTRs into RNA granules, which are competent for subsequent transport in dendrites.

### 2.4. Rgs4 RNA Structure Contributes to Protein Expression

In addition to RNA localization, RNA structure might also regulate translation through binding of Stau2. We therefore aimed to unravel the role of RNA structures in guiding posttranscriptional regulation, including protein expression. To assess whether the identified *Rgs4* RNA duplex conveys such regulation, we performed luciferase reporter assays in developing primary cortical neurons (Figure 4). Deletion of F1 (*Rgs4* F1Δ) caused a significant increase in luciferase activity compared to *Rgs4* F1 reporter expression (Figure 4A). However, we did not observe any significant change in reporter RNA levels between the three relevant *Rgs4* constructs, indicating that the observed changes occur post-transcriptionally (Figure 4B). Importantly, in contrast to deletion of F1, deleting another previously identified SRS in close proximity to the highly prevalent RNA structure (SRS3 in Figure 2A) did not cause any effect (Figure 4C), highlighting the relevance of the F1 hairpin identified by hiCLIP. In order to test, whether the observed effect is specific for the *Rgs4* RNA hairpin, we analyzed two other newly identified hairpins from *Mapt* and *RhoA* (Figure 1B and Appendix A). Deletion of the respective RNA hairpin in *Mapt* (*Mapt* 3′-UTRΔ) yielded a decrease in luciferase activity compared to *Mapt* 3′-UTR reporter expression (Appendix A). In contrast, there was no change in expression when the respective RNA hairpin in *RhoA* (*RhoA* 3′-UTRΔ) was deleted (Appendix A). Together, our results show that different Stau2-bound RNA hairpins result in distinct effects on protein expression in cortical neurons. This suggests that RNA secondary structures exert a specific impact on protein expression.

To rule out the possibility, that the *Rgs4* hairpin interferes with luciferase activity during translation, we complemented our results using a GFP reporter assay in mature neurons (15 DIV, Figure 4D). Cells were transduced with lentiviral particles expressing either a well-characterized Stau2 short hairpin RNA (shRNA) [32] or control shRNA (shNTC) and subsequently transiently transfected with GFP-*Rgs4* 3′-UTR constructs (*Rgs4* FL, *Rgs4* F1 and *Rgs4* F1) (Figure 4D). Please note that adding a complex 3′-UTR to GFP reporter constructs led to a reduction in reporter expression in neurons due to the high number of regulatory elements embedded in the 3′-end [18,19,33]. On top of this general drop in protein levels, Stau2 downregulation significantly reduced GFP expression of both *Rgs4* F1 and—to a lesser but still significant extent—*Rgs4* FL containing reporters. In contrast, in this experiment involving mature neurons (15 DIV) *Rgs4* F1Δ remained at control levels, again indicating that the structural motive deleted in this construct is important for Stau2 interaction (Figure 4D). Moreover, GFP reporter RNA levels remained similar between shNTC and shStau2 conditions arguing primarily for an effect at the protein level (data not shown).

## 3. Discussion

RNAs have the intrinsic ability to fold and to form complex secondary structures in vitro [34] and in vivo [35]. Moreover, both the GC content and the folding energy influence the association of mRNAs with polysomes [33] suggesting that RNA structures regulate translation. In line with this notion is the finding that the GC content and in turn the propensity of RNAs to form secondary structures determines their localization to Processing bodies (P-bodies) and consequently RNA storage and decay [36]. Therefore, it is plausible that RNA secondary structures serve as an additional code to regulate posttranscriptional gene expression. Supportive for this finding is a previous study showing that the RBP Stau1 binds complex structures to regulate the fate of its target RNAs [26].

Here, we characterized the binding specificity of Stau2 to complex RNA structure both in the rat brain and in primary hippocampal neurons. In particular for Stau2, addressing its binding specificity in vitro has been challenging (e.g., [21,23,37]). Its impact on mRNA and protein expression, however, is clearly restricted to a group of specific functionally related genes [19,38] as well as to specific regions within a transcript, e.g., the 3′-UTR as well as introns [19,20]. Hence, there must be signals embedded within the mRNA that guide Stau2 binding and regulation. By exploiting a battery of different approaches including hiCLIP, luciferase and GFP assays as well as single particle localization of RNA granules within dendrites, we show that the Stau2 bound hairpin of its target *Rgs4* [19] regulates Stau2 RNP assembly and RNA transport in neurons. Our data complement previous findings supporting a role of RNA structure in phase separation and RNP assembly [7,31]. Another aspect that is regulated by RNA structures is protein expression. We observed that deleting the F1 fragment resulted in increased luciferase activity while GFP reporter expression remained unaffected. This finding can be explained by differences in luciferase and GFP protein stability. Another, not mutually exclusive, explanation might be that the *Rgs4* 3′-UTR serves as a platform for modifying enzymes that control function of the encoded protein. A similar mechanism has been observed for CD47 [39]. Moreover, we observed that deleting the F1 fragment reduced the Stau2 mediated effect on GFP expression. However, it did not mimic the reduction in GFP levels observed for the FL 3′-UTR upon Stau2 depletion. This effect points towards additional elements in the 3′-UTR of *Rgs4* [40] that are affected by the deletion showing that 3′-UTRs are likely to be equipped with numerous sequence and structure motifs.

Based on our findings and in light of recent studies, it is tempting to speculate that RNA structures crucially influence the assembly behavior into RNA particles. Thereby, the RNA itself determines whether it will be excluded from particle assembly or not [7,41]. This autonomous role of RNA is in line with our finding that different RNA structures apparently play distinct roles on protein expression. Therefore, future (biochemical and/or structural) work will have to identify how different RNA secondary structures determine different functional outcomes. These approaches will have to be complemented with sophisticated computational approaches to unravel evolutionary conserved structure motifs and identify critical parameters such as folding energy and nucleotide spacing. This will ultimately allow us to define a structure code for RNP assembly.

Over recent years, numerous studies have characterized the assembly dynamics of various types of RNPs, predominantly of stress granules and P-bodies [42]. Much less is known about the underlying driving forces of transport granule assembly that is needed for the localization of RNAs to synapses in (distal) dendrites [5]. With our current study, we define a first complex RNA secondary structure that recruits the double-stranded RBP Stau2 [13], which leads to RNA granule assembly and subsequent transport into dendrites [5]. Due to the important role of Stau2 in synaptic plasticity and behavior [43,44], it is plausible that its association with target RNAs will be regulated by synaptic activity [18]. Whether this is mediated by the RBP itself through posttranslational modifications [3] as has been recently shown for the SMN complex [45] is not (yet) known. Another option is that it is the RNA target instead of the RBP that might be responsible for the synaptic activity mediated expression control. Similar to proteins, also RNAs become modified in neurons [46]. Interestingly, these modifications have been recently shown to promote phase separation of the RBP bound RNA [47]. Consequently, alterations in RNA modification are likely to change in response to synaptic activity and might therefore alter RNA structure to promote RNP assembly. While we have shown the importance of RNA structure for this process in neurons, future studies are clearly needed to elucidate the complex interplay between RBP, target RNAs, RNA structures and medications.

## 4. Materials and Methods

### 4.1. Plasmids

Generation of luciferase plasmids (psiCHECK-2 vector, Promega, Fitchburg, WI, USA) was performed by insertion of PCR amplified DNA fragments in between the stop codon of Renilla open reading frame and the poly A signal using the following primers (5′–3′): rat *Rgs4* mRNA (ENSRNOG00000002773) 3′-UTR (position 728–2919 nt) as described [19]; *Rgs4 F1* (position 1064–1535 nt) forward XhoI caagttactcgagacacttcatg; reverse SalI ccatagcgtcgacctcaatg; *Rgs4 F1Δ* (position 1064–1510 nt) forward NdeI tttttcatatgcatatacattctg; reverse SalI cgcgtcgacttaaacacaaagg.; *Mapt* mRNA (ENSRNOT00000006947.8) 3′-UTR (position 1216–5138 nt) forward XhoI aggcgatcgctcgagtcaggcccctggggc; reverse NotI gcggccagcggccgcaatcagagtaataactttat; *RhoA* mRNA (ENSRNOG00000050519) 3′-UTR (position 583–1584 nt); Site-directed mutagenesis was performed on psiCHECK-2 constructs to generate the *Stau2* duplex deletions through sequencial PCRs: for *Rgs4 F1Δ* fwd1:ctagtcgtcgtcgacttctc; rev1:caagtttcttaccaaaccaatgaaaacc; fwd2:ggttttcattggtttggtaagaaacttg; rev2:cagaatgtatatgcatatgttctatata. For *MaptΔ* fwd1:ggcgatcgctcgagtcaggcccc; rev1:ggcatgataggacaggggaccaggcttgag; fwd2:ctcaagcctggtcccctgtcctatcatgcc; rev2:gctctccctgctggtactagtgtccttttc. For *RhoA*Δ, fwd1:gcgatcgctcgagagccttgtgacg; rev1:gcagctgacagaaaatacacaaaagttaccaac; fwd2:gttggtaacttttgtgtattttctgtcagctgc; rev2:tttttactagtgaccccccaccc. The plasmids for pUBC-*NLS-ha-tdMCP-GFP* and pRSV-*LacZ-128xMS2* have been described [18]. pCMV-*eGFP*-STOP [20] (pEGFP-C1, Clontech, Mountain View, CA, USA) pRSV-LacZ-128xMS2 reporter plasmids were produced from psiCHECK-2 plasmids following the same procedure: pRSV-*LacZ-128xMS2* plasmid was linearized using XhoI/SmaI; pCMV-*eGFP*-STOP plasmid was linearized using SalI/XmaI; all inserts were extracted from the psiCHECK using NotI, filled with klenow and then cut once more with XhoI before ligation to linearized plasmids. The lentiviral packaging plasmids psPAX2 and pcDNA3.1-VSV-G have previously been described [19]. Lentiviral plasmids pFu3a-H1-sh-NTC-pCaMKIIα-tag-RFP and pFu3a-H1-sh-*Stau2*-2-pCaMKIIα-*tagRFP* were generated by exchanging the UBC promoter for the CaMKIIα promoter [18]. All enzymes were purchased form New England Biolabs (Ipswitch, MA, USA), oligonucleotides were purchased from Eurofins Genomics (Ebersberg, Germany).

### 4.2. Neuronal Cell Culture, Treatment, Transduction and Transfection

All animals in this study were used according to the German Welfare for Experimental Animals (LMU Munich, Government of Upper Bavaria). Rat hippocampal neuron cell cultures from embryos at day 17 (E17) of timed pregnant Sprague-Dawley rats (Charles River Laboratories, Wilmington, MA, USA) were generated as described previously [48]. Briefly, E17 hippocampi were dissected and trypsinized, cells dissociated and plated on poly-L-lysine coated coverslips and cultured in NMEM+B27 medium (Invitrogen, Carlsbad, CA, USA) with 5% CO_2_ at 37 °C. For cortical cultures, E17 cortices were trypsinized and dissociated, cell suspension sequentially filtered through 100-, 70-, and 40-μm cell strainers and then plated at a density of 100,000 cells/cm^2^ on poly-L-lysine coated 60 mm dishes. For biochemistry assays (protein/RNA luciferase assay, and EGFP RNA assay), cortical neurons were used due to the high amount of cells needed for these analyses, in order of minimize the number of animals needed. Neurons were transfected at 7 DIV (days in vitro) by calcium phosphate coprecipitation [48] and used 24 h after transfection. For Stau2 knock-down conditions, cortical neurons were transduced with lentiviral suspension at 4 DIV, transfected at 7 DIV and lysed at 8 DIV. For experiments involving MS2-RNA imaging, hippocampal neurons were routinely used as they represent the most reliable and well defined neuronal cell culture system (see for example [48]). Hippocampal neurons were transduced with lentiviral suspension at 10 DIV, followed by transient transfection via calcium phosphate coprecipitation at 14 DIV and fixation at 15 DIV.

### 4.3. Lentivirus Production

Lentiviral particles for shNTC and shStau2 were generated from HEK-293T cells cotransfected with psPAX2, pcDNA3.1-VSV-G and the respective pFu3a plasmids using calcium phosphate coprecipitation. After 48 h virus production, supernatants were filtered (0.45 µm PVDF Millex-HV; Millipore, Burlington, MA, USA), concentrated by ultracentrifugation (65,000× *g*, 140 min, SW 32 Ti rotor; Beckman Coulter, Brea, CA, USA) and resuspended in Opti-MEM™ (Life Technologies, Carlsbad, CA, USA) [19].

### 4.4. RNA Extraction, cDNA Synthesis and qPCR

RNA was isolated from 8 DIV cultured cortical neurons and P0, P7, P14 and P28 rat brain cortices using TRIzol according to the manufacturer’s manual (Invitrogen, Carlsbad, CA, USA). cDNA synthesis and qRT-PCR were performed as described [20]. The sequences of the primers were (5′–3′): *Ppia*_F: gtcaaccccaccgtgttctt; *Ppia*_R: ctgctgtctttggaactttg; *Rgs4*_F: agtcccaaggccaagaagat; *Rgs4*_R: aacatgttccggcttgtctc; *Stau2*_F: gaacatctcctgctgctgaag; *Stau2*_R: atccttgctaaatattccagttgt; *RRL*_R: acgtccacgacactctcagcat; *FFL*_F: gagtctatcctgctgcagcac; *FFL*_R: ctcgtccacgaacaccactc; *eGFP*_F: acccagtccgccctgagcaa; *eGFP*_R: gcggcggtcacgaactccag; *Kan*_F: tgcctgcttgccgaatatca; *Kan*_R: atatcacgggtagccaacgc.

*hiCLIP* was performed essentially as described in [26] with the same modifications in the general iCLIP procedure as described recently (Lee et al., bioRxiv, https://www.biorxiv.org/content/10.1101/2021.08.27.457890v1.full, accessed on 12 October 2020).

### 4.5. Bioinformatics

Data from Stau2 and input iCLIP experiments (performed in triplicate) were processed using the *nf-core/clipseq* pipeline (https://nf-core/clipseq, accessed on 12 October 2020) to identify crosslink positions. The Ensembl *Rattus norvegicus* Rnor 6.0 assembly with the v100 annotation release was used. As extended neuronal 3′-UTRs are unannotated, we extended this reference annotation with matched 3′-end sequencing data from these rat brains. Gene level quantification was performed, discarding crosslinks that overlapped more than one gene (equivalent to HTSeq’s “union” method). To identify genes with enriched Stau2 binding, we used the DESeq2 framework and the shorth function from genefilter (which performs better for low count data) to estimate size factors. Genes that had fewer than 5 normalized counts in 3 or more conditions were filtered out. We performed a differential analysis using the Wald test to compare IP and input conditions with an adjusted *p*-value threshold of 0.01 determining statistical significance.

Stau2 hiCLIP data were processed using our custom analysis pipeline *Tosca* which identifies hybrids from hiCLIP and proximity ligation experiments and visualized using IGV.

The overlap between distinct databases was performed using the online tool for Venn diagrams Venny (https://bioinfogp.cnb.csic.es/tools/venny/index.html, accessed on 12 October 2020) [49]. For GO enrichment analysis we used the DAVID web service [50]. First, we defined the overlapped list of genes as our interest list, and all expressed genes as background, retrieving the Functional Annotation Chart with *p*-value < 0.05, FDR < 5% and count threshold of 3.

### 4.6. Rgs4 RNA Secondary Structure Prediction

The thermodynamic structure prediction of the region corresponding to nucleotide position 336–807 of the 3′-UTR sequence of rat *Rgs4* mRNA (NM_017214.1) was predicted using the RNAfold server within the ViennaRNA web services (http://rna.tbi.univie.ac.at, accessed on 12 October 2020) [51]. Standard options were used, but no GU pairs at end of helices were allowed. For the visualization of the experimentally detected (hiCLIP) secondary structure, the two arms forming the stem (274,300 nt and 469–492 nt) were fused at nucleotide positions 300 and 469 into a single RNA strand and structure was predicted as described above.

### 4.7. Luciferase Assay

3′-UTR fragments of interest were cloned downstream of the *Renilla* luciferase gene into the psiCHECK-2 vector (Promega, Fitchburg, WI, USA). The empty Firefly luciferase reporter was used as control. Cortical neurons (100,000 cells per 24-well) were transfected with 8 µg of reporter plasmid (6 wells of a 24-well plate) using calcium transfection. Luciferase activity was measured 24 h after transfection using a Centro XS3 LB 960 High Sensitivity Microplate Luminometer (Berthold, Bad Wildbad, Germany). Ratios of *Renilla*/Firefly luciferase activity were calculated and normalized to the luciferase empty vector values. A minimum of 3–6 independent experiments were performed. The normalized values from independent experiments were used to determine significant differences using the student’s *t*-test.

### 4.8. Imaging-Based GFP Expression Assay in Primary Neurons

3′-UTR fragments were cloned downstream of the *GFP* ORF into the p*EGFP*-STOP-C1 vector under the control of CMV promoter. The p*EGFP*-STOP-C1 reporter plasmid was used as control. First, 10 DIV primary rat hippocampal neurons were transduced with lentivirus particles expressing shNTC or shStau2. The corresponding shRNA sequence in pSuperior (si-2-2) has been validated extensively [18,19,20,48]. Routinely, we achieve substantial downregulation of Stau2 in neurons (ca. 70–80%) upon transfection or transduction [18,38].

For regulator plasmid based transient transfections, we use 14 DIV rat hippocampal neurons and 3 μg of reporter plasmid or empty vector control (p*EGFP*-STOP-C1) using calcium phosphate and fixed at 15 DIV [52].

### 4.9. Imaging and Data Analysis

Images were acquired on a Zeiss Z1 Axio Observer microscope including a 63× Plan-Apochromat oil immersion objective (1.40 NA), HXP 120 C and Colibri light sources and the Axiocam 506 mono camera using Zeiss Zen software (Zeiss, Oberkochen, Germany). Average GFP cell body intensity was quantified using Zen. At least 20 transfected cells per experiment (*n* = 3) were used for quantification. For MS2 particle localization and Stau2 co-localization experiments, overview images, including phase contrast pictures, were taken at the dendritic plane and z-stacks of the whole cell was acquired (30 planes with 0.26 µm step-size). The z-stack images were deconvolved using the Zeiss Zen software deconvolution module with 50% of the constrained iterative method settings. For the analysis of MS2 localization in dendrites, a maximum intensity z-projection was performed in ImageJ (version 1.50e). For 128xMS2 quantification one dendrite per cell was selected and straightened using the segmented line tool with a width of 70-pixels. Particles were manually detected using the multipoint tool and the distance to the soma and number of particles to the soma was measured using a script written for ImageJ to extract the x position for each particle in μm. Data were presented as total number of particles per dendrite, number of particles per 40 µm dendritic segment (histogram with 40 µm bin width) and average distance of particles per dendrite. For all experiments, ≥15 dendrites per condition from four independent experiments were selected for quantification. For the analysis of co-localization events between tagRFP-Stau2^62^ and the MS2 reporter constructs, images were analyzed using ImageJ. One dendrite per cell was selected for co-localization analysis. A region of interest along the whole dendrite, 40 µm away from the cell body, was selected using the segmented line tool (40 pixel line-width). The total number of MS2 particles per dendrite was manually counted using the multipoint tool, followed by manual assessment of co-localization with tagRFP-Stau2 signal. The ratio of tagRFP-Stau2 positive MS2 particles over total number of MS2 particles per dendrite was calculated. For each condition and biological replicate, 15–30 cells were analyzed and image analysis was performed blind.

### 4.10. Statistics

Microsoft Excel, the R software (version 4.1.0) [53] and prism software (version 5 GraphPad, San Diego, CA, USA) were used for data processing, plotting and statistical analysis [54]. Figures represent mean ± standard error of the mean (SEM) of at least three independent biological replicates. Data was tested for normal distribution using Kolmogorov-Smirnov-Test. Asterisks represent *p*-values obtained by Kruskal-Wallis tests with Dunn’s multiple comparison, Mann-Whitney-test, one-way ANOVA with subsequent Tukesy’s multiple comparisons test, Kolmogorov-Smirnov test, one-sample *t*-test and either paired or unpaired two-sided student’s *t*-test using the mean values per experiment (* *p* < 0.05, ** *p* < 0.01, *** *p* < 0.001), as indicated.

## Figures and Tables

**Figure 1 ijms-22-13021-f001:**
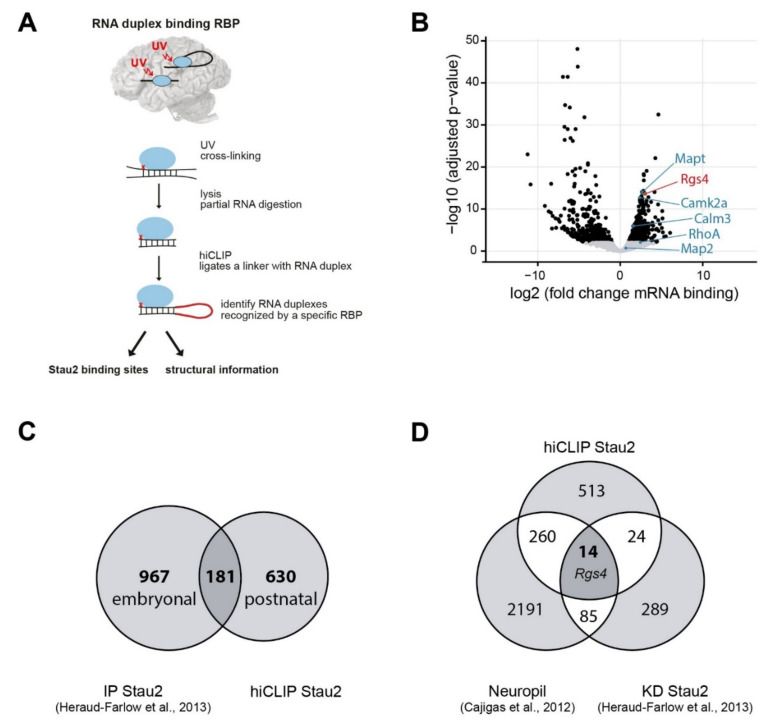
Stau2 RNA duplexes are enriched in dendritically localized transcripts with extended 3′-untranslated region (UTR). (**A**) Schematic overview of hiCLIP (RNA hybrid and individual-nucleotide resolution ultraviolet crosslinking and immunoprecipitation) in brain. (**B**) Volcano plot showing differential enrichment of Stau2 bound mRNA transcripts from postnatal rat brains (P35), comparing Stau2-immunoprecipitation (IP) with input. The previously identified Stau2 target, *Rgs4*, known to localize in dendrites, is highlighted in red. Examples of additional Stau2 targets are highlighted in blue: *Calm3*, *CamK2a*, *Mapt*, *Map2* and *RhoA*. (**C**) Venn diagram displaying the overlap between Stau2-IP mRNA targets from embryonic brain [19] and the newly identified Stau2-hiCLIP targets obtained from postnatal brains (P35). (**D**) Venn diagram displaying the overlap between Stau2 mRNA targets at P35, Stau2 up- and downregulated targets from embryonic brain [19] and neuropil-enriched mRNAs derived from hippocampal CA1 region [27].

**Figure 2 ijms-22-13021-f002:**
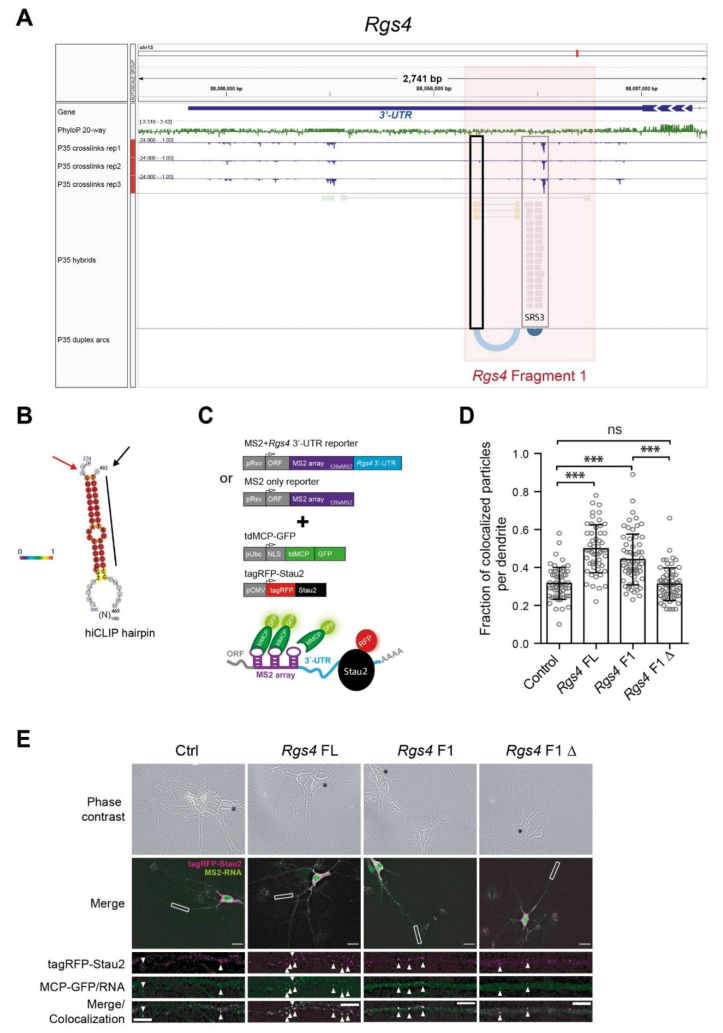
*Rgs4* RNA duplex is sufficient to drive Stau2-ribonucleoprotein particle (RNP) assembly. (**A**) Coverage of Stau2-hiCLIP (P35 hybrids) and iCLIP (P35 crosslinks) sites in *Rgs4* 3′-UTR (blue, in 3′- to 5′-orientation as shown in integrative genomic viewer (IGV), reverse strand) is depicted together with the PhyloP (100× vertebrates) sequence conservation (green) and the duplex linking arc plot. The 3′-UTR fragment 1 (pink shaded box) and fragment 1 deletion (black box) used in panels (**D**,**E**) are highlighted. (**B**) Display of the identified *Rgs4* RNA duplex. Please note that in silico folding of *Rgs4* F1 did not recapitulate the Stau2 bound hairpin (Appendix A). Arrows indicate the 5′- (red) and end 3′- (black) end of the structure, as well as the resulting loop size (*n* = 169). Grey nucleotides represent adjacent parts of the *Rgs4* 3′-UTR. The vertical black line indicates the 20 nt deletion arm within *Rgs4* F1. (**C**) Scheme of the control MS2 and MS2 + *Rgs4* 3′-UTR reporter constructs, the tdMCP-GFP and the tagRFP-Stau2^62KD^ expression cassettes and the Stau2/MS2 co-localization system. pRSV, Rous sarcoma virus promoter; pUBC, Ubiquitin C promoter; CMV, cytomegalovirus promoter; ORF, open reading frame; NLS, nuclear localization signal, tdMCP, tandem MS2 coat protein; UTR, untranslated region; GFP, green fluorescent protein; tagRFP, red fluorescent protein. (**D**) Quantification of the co-localization events between tagRFP-Stau2 and MS2 particles relative to total number of MS2 particles in dendrites, starting from 40 µm away from the cell body. Error bars are standard deviation from all dendrites (shown as individual dots) from three independent biological replicates; Kruskal-Wallis tests with Dunn’s multiple comparison were applied; asterisks represent *p*-values *** *p* < 0.001; ns, not significant. (**E**) Representative phase contrast and deconvolved fluorescent maximum z-projection images of hippocampal neurons transfected at 14 DIV with tagRFP-Stau2, tdMCP-eGFP and the indicated MS2-RNA reporters for 16 h. Scale bar, 20 µm. Dendrite magnification images are single z-planes. Asterisks indicate transfected cells, arrowheads indicate co-localization events. Scale bar, 5 µm. FL, full length; F1 fragment 1; F1Δ fragment 1 with a 20 nt deletion.

**Figure 3 ijms-22-13021-f003:**
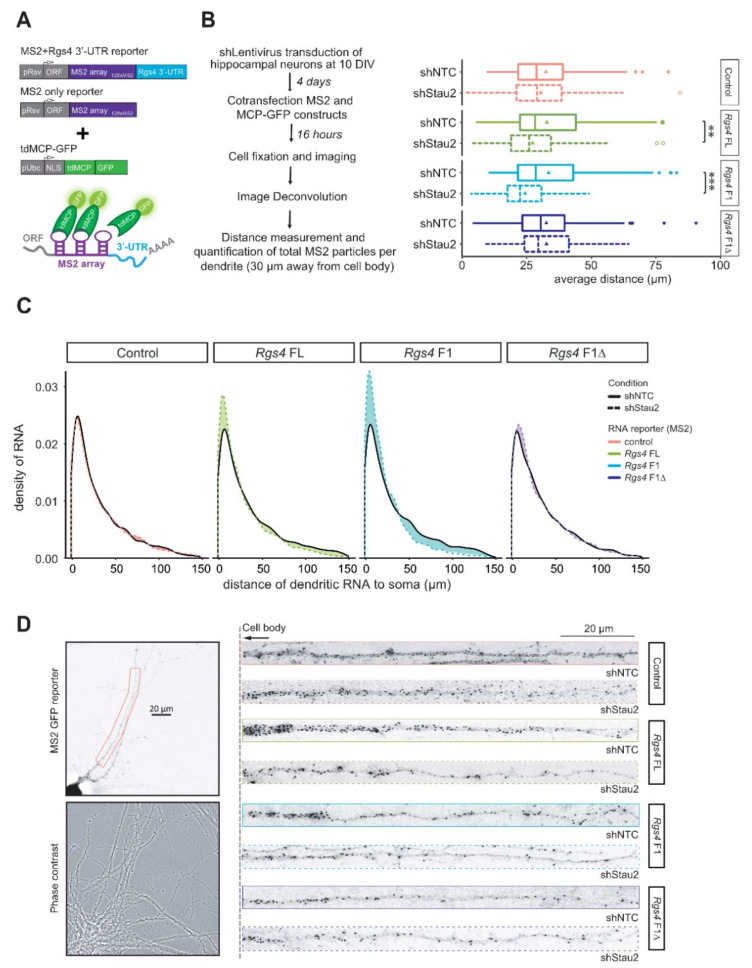
Stau2 mediates RNA structure dependent dendritic localization. (**A**) Scheme of the different MS2 + *Rgs4* reporter constructs (only *Rgs4* 3′-UTR displayed) including control MS2, *tdMCP-GFP* expression cassettes and the MS2 system. pRSV, Rous sarcoma virus promoter; pUBC, Ubiquitin C promoter; ORF, open reading frame; NLS, nuclear localization signal; tdMCP, tandem MS2 coat protein; UTR, untranslated region; GFP, green fluorescent protein. (**B**) Experimental setup (left) and boxplot (right) of the average distance (in µm) of MS2 particles per dendrite. Unpaired student’s *t*-test. DIV, days in vitro. Data from 4 independent biological replicates with at least 15 dendrites per replicate. Asterisks represent *p*-values (** *p* < 0.01, *** *p* < 0.001). (**C**) Density plot displaying the distance between MS2 particles in control (shNTC) and Stau2 KD conditions (shStau2) in co-transfected rat hippocampal neurons. Data represent the indicated MS2 + *Rgs4* reporter mRNAs or control MS2. Colored areas represent differences in RNA distribution in dendrites upon Stau2 downregulation. (**D**) Left: Phase contrast and 128xMS2 GFP fluorescence in a rat hippocampal neuron at 14+1 DIV expressing both tdMCP-GFP and 128xMS2 + *Rgs4* 3′-UTR reporter RNA. Boxed region indicates a representative distal dendrite. Right: Deconvolved straightened images of dendrites are shown expressing both tdMCP-GFP and 128xMS2+ control, *Rgs4* 3′-UTR full length (FL), *Rgs4* fragment 1 (F1) or *Rgs4* fragment 1 deletion (F1Δ), in control (shNTC) and Stau2 KD conditions (shStau2). Straightened images are cropped 30 µm away from the cell body (dotted line) for better particle visibility. FL, full length; F1 fragment 1; F1Δ fragment 1 with 20 nt deletion. Data from 4 independent biological replicates with at least 15 dendrites per replicate.

**Figure 4 ijms-22-13021-f004:**
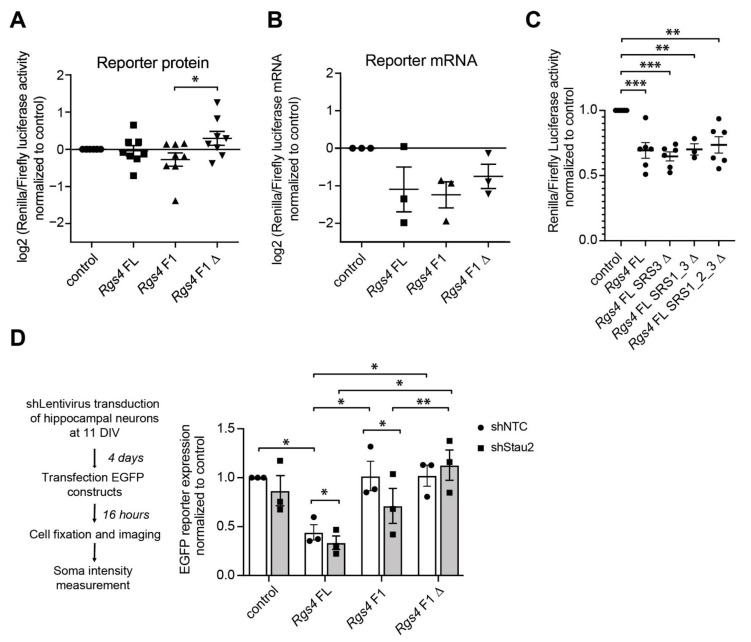
Lack of Stau2 significantly reduces GFP reporter expression in primary neurons. (**A**) Luciferase (RRL/FFL) activity in primary rat cortical neurons at 7+1 DIV transfected with *Rgs4* 3′-UTR constructs or control (no UTR). All data is normalized to control. (**B**) Luciferase *RRL* mRNA levels in primary rat cortical neurons at 7+1 DIV transfected with *Rgs4* 3′-UTR constructs or control (no UTR) normalized to *FFL* mRNA and to control. Symbols represent the individual conditions in (**A**,**B**). (**C**) Luciferase (RRL/FFL) activity in primary rat cortical neurons at 7+1 DIV transfected with the indicated control (no UTR) or *Rgs4 FL* constructs with the described SRS deletions (Δ) constructs [19]. All data is normalized to control. Unpaired student’s *t*-test. (**D**) Scheme of the experiment (left) and quantification of eGFP fluorescence intensity in the cell body of hippocampal neurons at 15 DIV, co-transfected at 14+1 DIV with the indicated *Rgs4* 3′-UTR or control (no UTR) constructs in Stau2 KD (shStau2) or mock conditions, respectively. Mann-Whitney-test (**A**), one-sample *t*-test (**B**), one-way ANOVA with subsequent Tukesy’s multiple comparisons test (**C**) as well as paired Student’s *t*-test (**D**). Error bars from (**A**,**C**) are SEM from at least 6 independent biological replicates. Error bars from (**B**,**D**) are SEM from ≥3 independent biological replicates. Asterisks represent *p*-values * *p* < 0.05, ** *p* < 0.01, *** *p* < 0.001. DIV days in vitro. FL, full length; F1 fragment 1; F1Δ fragment 1 with a 20 nt deletion.

## Data Availability

hiCLIP data are available via ArrayExpress (https://www.ebi.ac.uk/arrayexpress, accessed on 12 October 2020) under the accession code E-MTAB-11201.

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
