# Peer review of "RGS4 RNA Secondary Structure Mediates Staufen2 RNP Assembly in Neurons"

_ijms, 2021, doi:10.3390/ijms222313021_

Round 1

Reviewer 1 Report

RNA localization coupled to local translation has emerged as a key post-transcriptional mechanism underlying neuronal functions. While hundreds to thousands of mRNAs are known to localize to the dendrites of mature neurons, how these RNAs are selectively recognized by the transport machinery for proper localization and expression is still largely unclear. Bioinformatics studies have had difficulties identifying generic motifs/signatures promoting transcriptome-wide dendritic mRNA targeting. To date, only a very limited number of “zipcodes”, ie localization elements recognized by specific RNA binding proteins and essential for dendritic targeting, have been characterized. These zipcodes have been defined by specific sequence motifs, and the implication of dedicated RNA structures has so far been largely ignored.

In this study, the authors aimed at addressing this open question by performing a hiCLIP experiment to systematically identify RNA structures recognized by Staufen2, a dsRNA binding protein promoting the targeting of target mRNAs to dendrites. They characterized an RNA duplex found in the 3’UTR of the Rgs4 mRNA, showing that it is required for recruitment of Staufen2 in vivo, and for Staufen2-dependent localization and translational control.

This manuscript is overall well-written, data are of high quality and experiments performed with adapted controls. A number of points listed below should however be addressed before publication to i- better present the context of this study, ii- explain the relative behavior of control and Rgs4 3’UTR RNA constructs and iii- clarify the respective roles of Staufen2 and the RNA duplex in translational control. As this manuscript for the first time describes the Staufen2 hiCLIP dataset, it may also be good that the authors link their data to a public repository.

Major points:

1- The introduction is very short and does not provide the reader with a precise understanding of the context of this study. A description of Staufen2 general binding properties (dsRNA binding, lack of in vitro binding specificity…) would help the reader understand the bases of the study. Furthermore, explaining the specificity of Staufen1 and Staufen2 would better highlight the importance of studying Staufen2 binding properties. Last, highlighting the current lack of in vivo-validated Staufen2 binding elements would emphasize the need for studying such specific elements in living cells.

2- The manuscript starts with a description of the hiCLIP experiment initially performed by the authors to identify specific RNA structures bound by Staufen2. What is to be learned from this study regarding the structures bound by Staufen2 at the transcriptome-wide level is however not described in this manuscript. Furthermore, whether the authors have deposited their data in a public repository is unclear.

3- The exact position of the SRS3 with respect to the newly discovered RNA duplex is unclear.

4- line 177: “one possible explanation could be that Stau2 might also bind the control construct”. The fact that the control RNA construct localizes to dendrites as well as the construct containing Rgs4 3’UTR is puzzling. Although the images shown in Figure 1E fit with the hypothesis that Stau2 might bind to the Rgs4 coding sequence, the observation that Stau2 inactivation does not affect the localization of the control construct does not. Have the authors tried to perform smFISH experiments using ms2 probes to exclude the possibility that part of the MCP-GFP+ foci are not containing ms2-tagged RNAs? Alternatively, have the authors quantified the relative abundance of the different ms2-tagged RNAs under study (it may be that the levels of the control ms2-RNA are higher than those of the ms2-RgS4 3’UTR RNA and those that the fraction in dendrites is lower?)

5- Line 182 :” the identified RNA structure is essential for proper localization of Rgs4 mRNA into distal dendrites” This conclusion appears to be too strong.

6- Figure 4A and B:

- Given the heterogeneity of Rgs4 FL behavior, and the fact that Rgs4 FL likely contains regulatory elements subjected to Stau2-independent regulation, I would suggest the authors to only display the control, Rgs4 F1 and Rgs4 F1D conditions (similar for Figure 4D).

- The parametric tests used for statistical comparison of the different datasets are not adapted to sample sizes (same for supplementary Figure 4f).

- It would be easier if the scale of the y axes would be similar in Figure 4A and B.

7- Figure 4D: The authors should comment on why an increase is observed for the F1Delta construct (compared to the F1 construct) when using luciferase reporters (Fig 4A) but not when using gfp reporters (Fig 4D). Furthermore, why is it that disrupting the RNA duplex does not mimic inactivation of Staufen2 in Fig 4D?

Minor points :

- line 65 : the Sugimoto et al, 2015 reference is not properly formatted

- line 104 : “ To identify specific RNA structures, we filtered our dataset …”. The described filters used by the authors were not designed “to identify specific RNA structures”, but rather biologically relevant targets

- line 111 : “several members of this pathway are regulated by Stau2 (Figure 1D)”. It would be interesting to know the identity of the mentioned members; Figure 1D is not providing such information

- line 128 “is responsible for Stau2 assembly”. The meaning of “Stau2 assembly is unclear”.

- line 159, legend to Figure 2D: “one-way ANOVA and unpaired student’s t-test were applied”. Why a t-test was used for this graph is unclear

- Figure S3B: The number of MS2 particles per dendrite decreases upon Stau2 inhibition in all conditions tested. Is it linked to a reduction in the length of dendrites upon shStau2 treatment?

Author Response

see word file, referee #1

Reviewer 2 Report

In this manuscript by Fernandez-Moya et. al., the authors present interesting findings on roles for mRNA secondary structures in RNA binding protein interaction. They performed hiCLIP followed by Stau2 RNA immunoprecipitation (RIP) from P35 rat brain lysates. Further they compared their hiCLIP-RIP-Seq data set with published literatures and narrowed down to Rgs4 mRNA as the top Stau2 interactor. With hiCLIP they identified the 167 nts sequence (F1) present in the upstream 3'UTR region of Rgs4, which is responsible for Stau2 interaction. By performing deletion analysis of this sequence combined with MS2-reporter, the authors showed that in absence of the 167 nt sequences (F1Δ);

  • The Rgs4 reporter co-localizes significantly less with Stau2.
  • The Rgs4 reporter localizes significantly less to dendrites.
  • and the Rgs4 reporter translates at a higher rate. 

Under Stau2 knockdown conditions;

  • Dendritic localization of Rgs4 reporter mRNAs carrying  FL 3'UTR or F1 3'UTR were significantly reduced but dendritic transport of F1Δ was not  dependent on Stau2. 
  • Reduction of Stau2 protein in Hippocampal neurons reduced expression of Rgs4 reporters carrying  FL 3'UTR or F1 3'UTR but not F1Δ. 

Major Comments;

Roles of mRNA secondary structures in RNA granule formation and mRNA transport is a very interesting area of research and the authors have rightly mentioned that this area if research has not been explored extensively. My concern is while the authors make the title of the article very broad w.r.t. RNA   secondary structures regulating Stau2 RNP assemblies, they show data only for one mRNA. So I think the title of the manuscript needs to be amended to reflect the data presented in the article. Moreover after identifying the F1 region in Fig. 1, the authors never tested the how deletion of the F1 region affected binding of the F1Δ reporter mRNA with Stau2.  

Specific Comments;

  1. Sentence 143-144: This experiment only shows colocalization of reporter mRNA with Stau2 is affected. So conclusion on binding should be avoided. 
  2. Sentence 143-144: Would recommend to use "....occur post-transcriptionally".
  3. Keeping the data in Fig. 4A in mind, I was surprised to see that the EGFP- F1Δ reporter under control sh condition is not higher than the control condition In Fig. 4D. The authors should explain this discrepancy. 

Author Response

see rebuttal, referee #2

Reviewer 3 Report

In this Ms. the Authors present "RNA Secondary Structure Mediates Staufen2 RNP Assembly in

Neurons". In this context, the Authors based their research on the recent literature of Stau1 and the hiCLIP method to elucidate how the structure (specifically secondary) of RNA influences the RNP assembly mechanisms, suggesting a strong dependence of the protein-RNA ensemble on a hairpin with a bulge obtained via hiCLIP among other methods. In particular, the authors based their study in unveiling the role of this particular hairpin in the whole Stau2 RNA assembly, which is at a molecular level a current hot-topic and biologically speaking very relevant. 

This work could inspire tractable and rigorous in-silico studies, where the focus will be the tertiary structure of RNA and their corresponding internal energies and conformational response, which may provide a robust interpretation of what only by experimental means is sometimes tedious to elucidate. I recommend attending a couple of remarks and comments that should be clarified before further rounds for publication:

  1. In Fig 2, line 152: The caption refers to Figure S1 A, which is not the correct reference. Moreover  in Fig 2B: What is the probability of the white/grey depicted nucleotides?

  1. In general, when RNA secondary structure is tackled, other shape and form variables need to be considered, as for example the end-2-end distance is a typical parameter obtained for mRNA fragments. In addition, if ViennaRNA has been used for the analysis perhaps a comparison between different MLD (maximum ladder distances) can be shown.

  1. In Fig. 3: The average distances tackled/measured seem to be at a completely different scale than the secondary structure shown in Fig2 B. A discussion on the effects from a couple of nm in um scales would improve an interpretation of the results.

  1. In Fig S2 A: The sketched secondary structure is not citing from which in-silico study it is coming from, nor details on how it was obtained are given.

  1. Discussion: The Authors introduce a few concepts that are not necessarily matching this study, like the folding energy. Perhaps the outlook may focus on joining experiments with theoretical predictions to give a fundamental interpretation of why such specific secondary structure is so relevant for the RNP assembly.

  1. Within the main text introduction different RNA secondary structure critical contributions to regulate RNA function are named. However, the modeling of RNA structure prediction could also improve the molecular understanding and mechanisms described in this manuscript. I recommend to mention the importance of the RNA structure secondary and tertiary like the one done with several RNA prediction models which analyse shape, energies and help building an idea of the relevance of secondary structures in this molecule, an interesting work in this direction is: https://doi.org/10.3390/v13081555.

Author Response

see rebuttal, referee #3

Round 2

Reviewer 1 Report

The authors have addressed most of my questions/comments.

The only minor modifications to be made relates to the graphs shown in Figures 4A and 4B (indications of statistical significance got lost in the revision process).

Author Response

See new rebuttal

Reviewer 2 Report

The authors have substantially addressed questions raised by me and other reviewers. 

I was referring to the "Sentence 201-201" to change "occur mainly at the protein level" to "occur post-transcriptionally". Sorry for the typing error. 

While I understand the concerns raised by the authors regrading Stau2's non-specific binding to RNAs in in vitro set ups, and removal of hairpin not completely abolishing Stau2 association with F1Δ reporter mRNA but I still feel this experiment is highly necessary for the conclusions this manuscript is making. Although F1 deletion did not affect Stau2 association completely but it brought it back to the control groups suggesting this experiment can very well be tested biochemically. I would strongly suggest the authors to perform Stau2 RNA Immunoprecipitaiton and look for interaction with WT vs. F1Δ reporter mRNA.  

Author Response

See new rebuttal

Reviewer 3 Report

Publish as it is.

Author Response

See new rebuttal
